# Large Stokes shift fluorescence activation in an RNA aptamer by intermolecular proton transfer to guanine

Mateusz Mieczkowski[1,2,3,7], Christian Steinmetzger[1,5,7], Irene Bessi[1], Ann-Kathrin Lenz[1], Alexander Schmiedel[1], Marco Holzapfel[1], Christoph Lambert [1,4 ✉], Vladimir Pena[3,6 ✉] & Claudia Höbartner [1,2,4 ✉]

Fluorogenic RNA aptamers are synthetic functional RNAs that specifically bind and activate conditional fluorophores. The Chili RNA aptamer mimics large Stokes shift fluorescent proteins and exhibits high affinity for 3,5-dimethoxy-4-hydroxybenzylidene imidazolone (DMHBI) derivatives to elicit green or red fluorescence emission. Here, we elucidate the structural and mechanistic basis of fluorescence activation by crystallography and time-resolved optical spectroscopy. Two co-crystal structures of the Chili RNA with positively charged DMHBO$^+$ and DMHBI$^+$ ligands revealed a G-quadruplex and a *trans*-sugar-sugar edge G:G base pair that immobilize the ligand by π-π stacking. A Watson-Crick G:C base pair in the fluorophore binding site establishes a short hydrogen bond between the N7 of guanine and the phenolic OH of the ligand. Ultrafast excited state proton transfer (ESPT) from the neutral chromophore to the RNA was found with a time constant of 130 fs and revealed the mode of action of the large Stokes shift fluorogenic RNA aptamer.

[1] Julius-Maximilians-Universität Würzburg, Institut für Organische Chemie, Am Hubland, 97074 Würzburg, Germany. [2] Georg August University School of Science, GGNB Doctoral Program Biomolecules, Justus-von-Liebig-Weg 11, 37077 Göttingen, Germany. [3] Max Planck Institute for Biophysical Chemistry, Am Fassberg 11, 37077 Göttingen, Germany. [4] Center for Nanosystems Chemistry, Universität Würzburg, Am Hubland, Würzburg, Germany. [5] Present address: Karolinska Institute, Stockholm, Sweden. [6] Present address: Institute of Cancer Research (ICR), London, UK. [7] These authors contributed equally: Mateusz Mieczkowski, Christian Steinmetzger. ✉email: christoph.lambert@uni-wuerzburg.de; vlad.pena@icr.ac.uk; claudia.hoebartner@uni-wuerzburg.de

Fluorogenic aptamers specifically activate the fluorescence of conditional fluorophores for RNA imaging and biosensor applications in vitro and in vivo[1–3]. Prominent examples of fluorogenic aptamers include the malachite green aptamer[4], Spinach[5], Broccoli[6], Corn[7], Chili[8], Mango[9], SiRA[10], Coral[11], and Pepper[12] aptamers, which use diverse classes of chromophores that are non-emissive in their free state but display strong fluorescence enhancement upon formation of the aptamer–ligand complex. The class of Spinach, Broccoli, Corn, and Chili are RNA mimics of green fluorescent proteins (GFP) that bind and activate derivatives of 4-hydroxybenzylidene imidazolone (HBI), a small-molecule analog of the tripeptide chromophore in GFP[13]. Spinach and Broccoli bind the difluorinated ligand DFHBI and its derivatives[5,14,15], and emulate the deprotonated chromophore and small Stokes shift of the enhanced GFP (eGFP)[16]. The dimeric Corn aptamer activates DFHO[7], a spectrally shifted oxime derivative of DFHBI, which resembles the chromophore of the red fluorescent protein DsRed[17]. Chili, on the other hand, is a large Stokes shift fluorogenic RNA aptamer, which was engineered from the dimethoxy-HBI (DMHBI)-binding 13-2 RNA aptamer[5] by truncation and sequence optimization. The resulting 52 nt Chili RNA preferentially binds and activates the positively charged chromophores DMHBI$^+$ and DMHBO$^+$ (Fig. 1a, b)[8,18] and mimics large Stokes shift fluorescent proteins[19]. In contrast to Spinach, which binds the deprotonated form of DFHBI, Chili exclusively binds the phenol form of DMHBI derivatives but enhances fluorescence of the phenolate forms. The positively charged groups in DMHBI$^+$ and DMHBO$^+$ substantially enhanced the binding affinity and fluorescence brightness[8,18].

Understanding the details of fluorescence activation requires knowledge of the underlying three-dimensional RNA structures. The co-crystal structures of Spinach and Corn aptamers in complex with their cognate ligands revealed the presence of G-quadruplex domains that stabilize the fluorophores by stacking interactions[1]. The binding sites feature complex quadruplex topologies[20–22] including the formation of a homodimer interface in Corn[7]. G-quadruplexes have also been identified in the Mango aptamer family, which binds and activates the thiazole orange conjugate TO1-Biotin[23–25]. Functional characterization of the Chili RNA has indicated that a G-quadruplex is likely involved in the RNA–ligand interaction[8,18], consistent with earlier hints obtained for the 13-2 aptamer with DFHBI[20]. However, the structural and mechanistic basis for ligand binding and deprotonation that cause the large Stokes shift in the fluorescence emission of the Chili aptamer have so far remained elusive.

Here we report the co-crystal structures of Chili bound to DMHBO$^+$ and DMHBI$^+$ at a resolution of 2.25 and 2.95 Å, respectively. A two-tiered G-quadruplex forms the core of the binding site and supports the ligand, which is anchored by an interaction of the oxime moiety with the RNA backbone. The methoxyphenol binding site is stabilized by metal ion coordination and by a short hydrogen bond between the OH of the ligand and the N7 of a guanine residue in an unpredicted Watson–Crick base pair. The structure suggests a model for the excited state proton transfer (ESPT) that is supported by time-resolved spectroscopy. These insights enabled structure-guided miniaturization of the Chili aptamer and will aid future engineering of fluorogenic modules for sensors and imaging applications.

## Results

**Overall structure of the Chili RNA aptamer bound to DMHBO$^+$ and DMHBI$^+$.** The 52-nt Chili RNA aptamer was co-crystallized with two of its cognate ligands, DMHBO$^+$ and DMHBI$^+$ (Fig. 1a, b). Crystals were grown by hanging-drop vapor diffusion and imaged under a fluorescence microscope (Fig. 1c). Crystals were derivatized with iridium(III) hexammine (by soaking or co-crystallization), experimental phases were obtained by single-wavelength anomalous dispersion (SAD), and the initial models were used for molecular

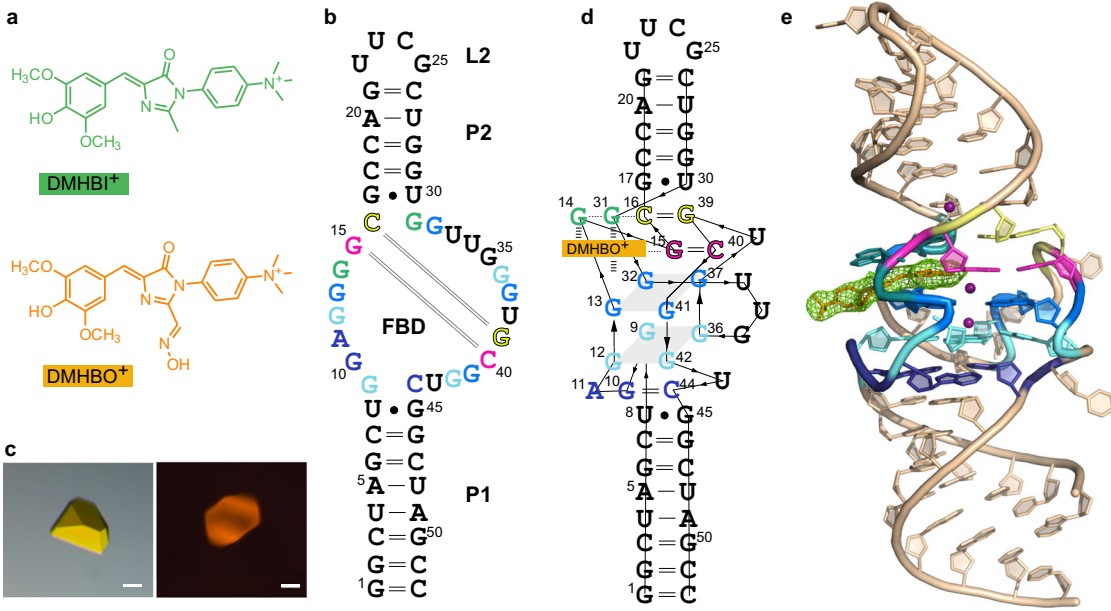

**Fig. 1 Overall structure of the Chili–DMHBO$^+$ and Chili–DMHBI$^+$ complexes. a** Chemical structures of DMHBO$^+$ and DMHBI$^+$. **b** Sequence and secondary structure of the 52-nt Chili RNA. P paired region, L loop, FBD fluorophore binding domain. Nucleotides shown in the same color interact by H-bonding in base pairs (green, pink, and yellow), base triple (dark blue), or G-quartets (light blue and cyan). **c** Images of representative Chili–DMHBO$^+$ crystals under white light (left) and UV irradiation with a mercury vapor lamp and fluorescence filter set (right). Scale bar: 20 μm. Images of $n = 15$ crystals were taken with similar results. **d** Tertiary fold diagram of Chili–DMHBO$^+$. Thin lines with embedded arrows indicate 5′−3′ connectivity. **e** Crystal structure of the Chili–DMHBO$^+$ complex. Color code of nucleotides in the FBD as in panels **b** and **d**. The green mesh represents the polder omit map of the DMHBO$^+$ ligand contoured at 4σ.

replacement (MR) with native datasets. The structure of Chili–DMHBO$^+$ was refined at a resolution of 2.25 Å. Likewise, the structure of Chili–DMHBI$^+$ was solved by MR and refined at a resolution of 2.95 Å (Supplementary Table 1). The unbiased electron density map allowed unambiguous tracing of the RNA and revealed the location of the fluorophore in each sample. The crystallographic asymmetric unit (ASU) of the native datasets contained four copies of RNA-ligand complexes (Supplementary Fig. 1), which were super-imposable with a root mean square deviation (rmsd) of 0.3 Å. The structural features of the DMHBO$^+$ and DMHBI$^+$ complexes are highly similar (Supplementary Fig. 2).

The Chili RNA folds into a single coaxial helical stack with a length of ~70 Å that contains two A-form duplexes P1 and P2 that are separated by the central fluorophore binding domain (FBD). The basal stem P1 contains 8 bp (nt 1–8 and 45–52), and the apical 5 bp stem P2 (nt 17–21 and 26–30) is closed by the UUCG tetraloop L2 (nt 22–25). The FBD spans nt 9–16 and 31–44 and accommodates the ligand binding site (Fig. 1b, d, e). P1 and P2 each contain a terminal G:U Wobble base pair (U8: G45 and G17:U30, respectively) flanking the FBD. The P1 stem transitions via a base triple into a two-tiered G-quadruplex, which constitutes the core of the FBD. The base triple is composed of G10:C44:A11, in which the Hoogsteen edge of A11 contacts the minor groove sugar edge of the cis Watson–Crick G10:C44 base pair, overall forming six hydrogen bonds. The G-quadruplex is formed by quartet T1 with guanines G9/G12/G36/G42, and quartet T2 with guanines G13/G32/G37/G41, and is stabilized by a central K$^+$ ion. The trinucleotide loop U33-U34-G35 is well-resolved in one of the copies in the ASU and partially disordered in the others. The ligand is immobilized by π-stacking between an unusual G14:G31 base pair and the T2 quartet. The latter also provides the platform for stacking of the long-range canonical Watson–Crick base pair G15:C40, which is followed by C16:G39 to elongate the stem P2 (Fig. 1d). The G17:U30 wobble base-pair provides the binding site for iridium hexammine in the major groove of the apical A-form helix[26]. The tetraloop L2 adopts a conventional UNCG fold[27], in which G25 is in syn conformation and the nucleobase forms H-bonds with the ribose edge of U22. Electrostatic interactions of the phosphate backbone of C24 and G25 in L2 with the positively charged trimethylammoniumphenyl side chain of a ligand bound to a neighboring Chili molecule serve as register for the head-to-head orientation of two copies of the RNA-ligand complex in the ASU. An intermolecular H-bond between the 2′-OH of G25 with N7 of G14 in the neighboring molecule constitutes an additional inter-subunit contact in the crystal lattice (Supplementary Fig. 3).

**Organization of the G-quadruplex core of Chili**. The architecture of the two-tiered G-quadruplex core in the Chili aptamer is characterized by three consecutive guanine steps (G12-G13, G36-G37, and G42-G41) with mixed parallel and antiparallel strand orientation, and one nonconsecutive (G9/G32) edge (Figs. 1d, 2a). All of the guanines in the G-quartets are in anti conformation (torsion angle $\chi = -120$ to $-160°$) with the exception of G32, which has high anti conformation (torsion angle $\chi = -79°$). The G-quartets T1 and T2 have inverted polarity featuring a partial overlap of the guanine imidazole rings (Fig. 2b, c). The G-quartets of the ligand binding site in the Chili RNA are stabilized by a central octacoordinate K$^+$ ion (M$_A$, Fig. 2c) with average K$^+$–guanine-O6 distances of 2.8 Å for T1 and 3.0 Å for T2. Notably, above the G-quadruplex and along the same central axis, a second K$^+$ ion (M$_B$, Fig. 2d) is coordinated to T2 (average K$^+$–guanine-O6 distance of 2.8 Å), the phenolic hydroxy and one methoxy group of the ligand (2.8 Å) as well as O6 of G15 (3.1 Å). The inter-K$^+$ distance is 3.5 Å.

**Architecture of the chromophore binding site**. The DMHBO$^+$ ligand adopts a nonplanar conformation within the binding site with twist and tilt angles of $\varphi = -27°$ and $\tau = 11°$, respectively (Supplementary Fig. 4). The deviation from planarity is primarily dictated by the stacking interactions with the RNA nucleobases. The benzylidene moiety of the ligand stacks on top of G13:G32 of the T2 quartet and G31 in the direction of the apical stem. The imidazolone moiety projects outward from the center of the quadruplex where it is anchored by stacking with G14 (Fig. 2d, e). The G14:G31 base pair is stabilized by a trans sugar-sugar edge (tSS) interaction that is mediated by mutual hydrogen bonds between the NH$_2$ of either guanine and N3/2′-O of the other guanine nucleotide. G31 is further held in place by a hydrogen bond between its own O6 and N4 of C16, which itself is part of a long-range Watson–Crick base pair with G39 (Fig. 2e). The cationic side chain of the ligand is located outside the perimeter of the binding pocket and is twisted at an angle of 60° with respect to the plane of the imidazolone ring (Fig. 2d, f).

Besides stacking interactions, the ligands establish polar interactions with the RNA (Fig. 2d). The oxime substituent of DMHBO$^+$ forms two hydrogen bonds with the phosphate backbone at G10 and the 2′-OH group of G9 (Fig. 2f). These contacts explain the difference in binding affinity between DMHBO$^+$ ($K_D = 12$ nM) and DMHBI$^+$ ($K_D = 65$ nM), and are likely responsible for the higher thermal stability of the fluorescent Chili–DMHBO$^+$ complex compared to the Chili–DMHBI$^+$ complex (see fluorescence melting curves in Supplementary Fig. 5).

**Identification of the proton acceptor**. In the crystal structure, the phenolic moiety of the ligand is coplanar with the long-range Watson–Crick base pair G15:C40 and forms polar contacts with the metal ion M$_B$ via the phenolic hydroxy group and one of the methoxy groups. Most notably, the phenolic hydroxy group comes into close contact with N7 of G15 (2.5 Å O–N distance, Fig. 2d), indicative of a mechanism that governs the protonation state underlying the fluorescent properties of the Chili-DMHBO$^+$ complex.

At pH 7.5 and in the absence of the Chili RNA, ~80% of DMHBO$^+$ is in the phenolate form exhibiting an absorption maximum at 555 nm (the p$K_a$ for the phenol/phenolate equilibrium of DMHBO$^+$ is 6.9, Supplementary Fig. 6.)[8] Upon binding of DMHBO$^+$ to the Chili RNA, the UV/Vis spectrum showed only an absorption maximum of 456 nm, indicating a shift of the equilibrium to the protonated phenol form upon formation of the RNA–ligand complex. The selectivity of the folded RNA aptamer for the protonated form of DMHBO$^+$ can be visually observed as a temperature-dependent colour change from purple to yellow upon cooling of the sample (Fig. 3b, c). This process is reversible and upon thermal melting of the RNA structure, DMHBO$^+$ is released, reverting the color change. In the absence of RNA, there is no temperature-dependent change in the phenol/phenolate equilibrium, and the absorption maximum of the free phenol form is 20 nm blue-shifted compared to the bound state. Upon excitation at 456 nm, steady-state fluorescence emission of the Chili–DMHBO$^+$ complex was observed exclusively from the phenolate form of the ligand, giving rise to the large Stokes shift of 136 nm (4800 cm$^{-1}$).

The same magnitude of the Stokes shift was observed when the G-quadruplex-stabilizing K$^+$ was replaced by Tl$^+$ or by Na$^+$ (Supplementary Fig. 7). With Tl$^+$, the fluorescence intensity was preserved, while Na$^+$ gave only 25% residual intensity, which was partially recovered by supplementing Na$^+$ with divalent Mg$^{2+}$. In contrast, the addition of Sr$^{2+}$ or Ba$^{2+}$ caused a bathochromic shift of both excitation and emission maxima by 16–18 nm only in the presence of Na$^+$. The same trend was observed for Chili–

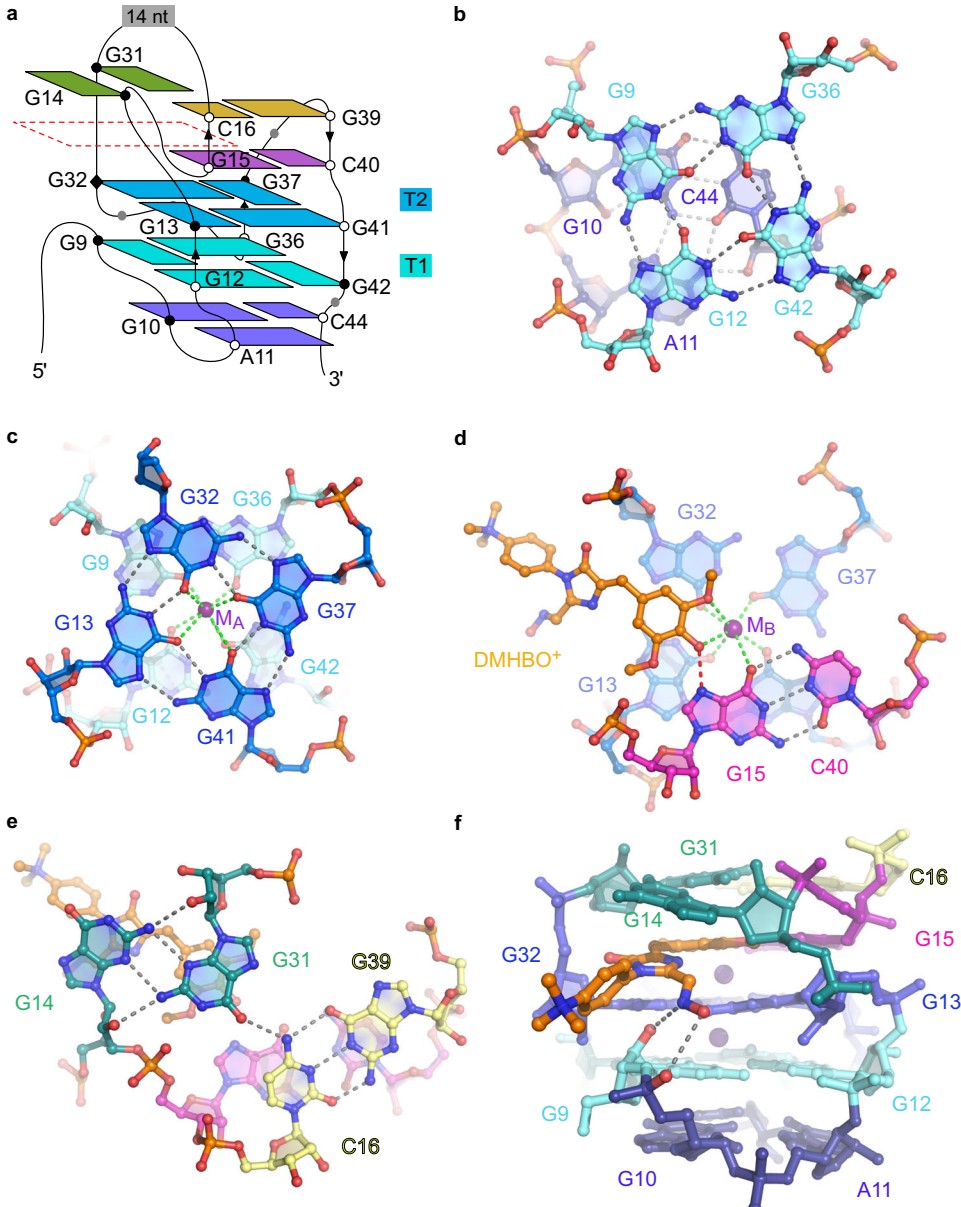

**Fig. 2 Organization of the G-quadruplex and ligand binding site of the Chili aptamer. a** Schematic representation of the connectivity and stereochemistry in the Chili aptamer core. White and black circles denote C3′-*endo* and C2′-*endo* ribose puckers, respectively. Black outline denotes *anti* conformation (all G's in T1 and T2 are *anti*, except G32, which is high *anti*, as denoted by black diamond). The position of the ligand is outlined with the red dashed line. The color code of the nucleobases is as in Fig. 1 and is also used in panels **b**–**f**. **b** G-quartet T1 and the G10:C44:A11 base triple. **c** T1 and T2 with the central potassium ion $M_A$, showing opposite polarity of the quartets with partial 5-5 stacking geometry. Gray and green dashed lines represent hydrogen-bonding and inner-sphere cation coordination, respectively. **d** The DMHBO$^+$ ligand is stacked on G-quartet T2 and coordinated with potassium ion $M_B$ and the G15:C40 base pair. The red dashed line represents a H-bond from the hydroxy group of the ligand to N7 of G15. **e** A *trans* sugar-sugar edge (tSS) base pair of G14 and G31 stabilizes DMHBO$^+$ via π-stacking interactions. O6 of G31 forms an additional hydrogen bond with the amino group of C16 in the C16:G39 base pair. **f** Stick representation of the ligand binding site showing hydrogen bonding interactions between the oxime moiety of the ligand and the RNA backbone.

DMHBI$^+$, which showed 6–8 nm blue-shifted emission maxima with Na$^+$/Sr$^{2+}$ and Na$^+$/Ba$^{2+}$, but not when the K$^+$ solution was supplemented with the heavier divalent metal ions Sr$^{2+}$, Ba$^{2+}$, or Mn$^{2+}$ (Supplementary Fig. 8)[18]. These results suggest that Na$^+$ is less tightly bound than K$^+$ and can be replaced by divalent metal ions to modulate the electronic coupling with the RNA-ligand complex.

In addition to ligand coordination, metal ion $M_B$ contacts the G15:C40 base pair in the FBD. Preliminary 2D NMR experiments of the Chili–DMHBI$^+$ complex in K$^+$/Mg$^{2+}$ solution revealed an unusual signal at a proton chemical shift of 8.6 ppm and $^{15}$N chemical shift of 142 ppm (Supplementary Fig. 9). The resonance unambiguously arises from an exchangeable proton of a guanine but does not belong to a shifted N1 imino proton of a canonical G:C Watson–Crick base pair. Together with the crystal structure, these results suggest that N7 of G15 participates in an ESPT mechanism.

To confirm the involvement of N7 of G15 in ligand binding and ESPT, we performed atomic mutagenesis and prepared a Chili RNA mutant containing 7-deazaguanine (c7G) at position

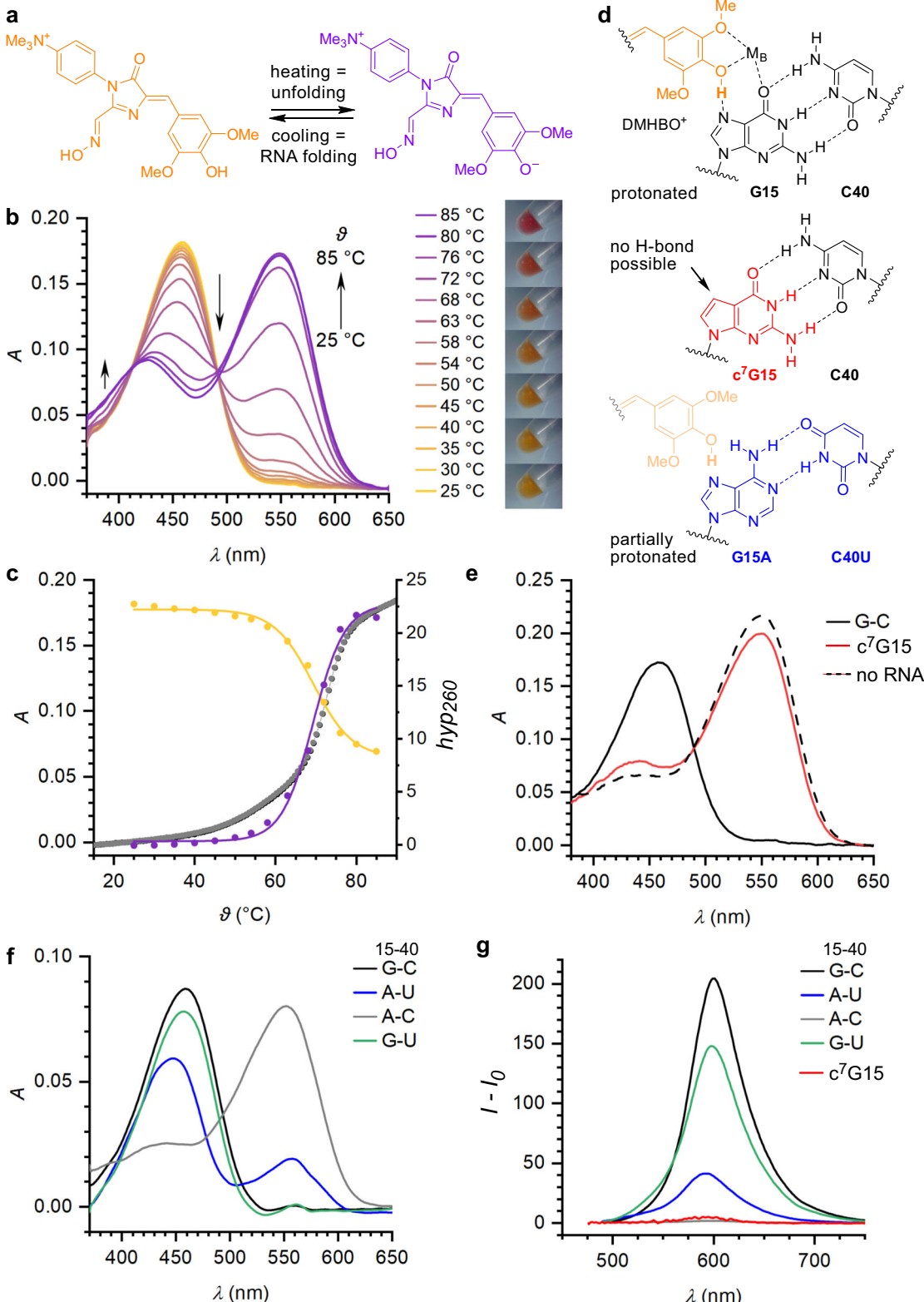

15. Removal of the proposed proton acceptor abrogated the ability to stabilize the protonated DMHBO+ ligand and resulted in complete loss of fluorescence activation (Fig. 3d, e, g). To confirm this result, the recently described fluorescent 4-cyanoindol ribonucleoside (r4CI) was incorporated at positions 34 or 46 of the Chili RNA, where it reported on ligand binding by serving as a FRET donor to DMHBO+[28]. No reduction in donor emission due to FRET was observed in c7G15 mutants of r4CI-modified Chili RNA, indicating that N7 of G15 is strictly required for ligand binding (Supplementary Fig. 10).

The role of the G15:C40 base pair was further examined by site-directed mutagenesis. The C40U mutation changed the Watson–Crick base pair into a Wobble base pair, resulting in 30% reduced fluorescence compared to the parent Chili–DMHBO+

**Fig. 3 Chili binds the phenol form of DMHBO$^+$ via H-bonding to N7 of G15. a** The phenol/phenolate equilibrium of DMHBO$^+$ allows monitoring of RNA folding. **b** Temperature-dependent absorbance spectra of the parent Chili RNA–DMHBO$^+$ complex (10 μM) and white light images of 100 μM solutions upon cooling from 85 °C to room temperature. **c** Absorbance (A) at 450 nm (yellow) and 550 nm (purple) as a function of temperature. The inflection point at 70 °C fits well to the reversible thermal melting transition of the RNA monitored at 260 nm (right axis: hyperchromicity ($hyp_{260}$), gray). **d** H-bonding between the ligand and N7 of G15 in wt-Chili and selected mutations. **e** c$^7$G15-modified Chili RNA is not able to shift the phenol/phenolate equilibrium. Absorbance spectra of the parent complex (solid black line) in comparison to free ligand (dashed black line) and with c$^7$G RNA mutant (solid red line). **f** Absorbance and **g** fluorescence spectra of Chili–DMHBO$^+$ complexes containing point mutations. The C40U mutation (Wobble base pair) is tolerated, while G15A completely disrupts ligand binding and fluorescence. Compensatory mutations of G15:C40 to A15:U40 only partially restore the fluorogenic binding site and result in a blue-shifted absorption maximum. RNA and ligand concentrations are 10 μM in **b** and **e** and 5 μM in **f** and **g**.

complex (Fig. 3g). The G15A mutation resulted in an A:C mismatch that destroyed the ligand binding site. The compensatory double mutant G15A and C40U was able to bind the ligand but displayed incomplete proton transfer (both before and after excitation), and a 10 nm blue-shifted absorption maximum of the protonated fraction (Fig. 3d, f). These results strongly support the functional role of the purine Hoogsteen edge as a general base for the proton transfer.

**Time-resolved optical spectroscopy.** Time-correlated single photon counting (TCSPC) on the ns timescale revealed multiexponential decays for both Chili–DMHBO$^+$ and Chili–DMHBI$^+$ complexes but did not resolve the kinetics of proton transfer (Supplementary Fig. 11). Therefore, the photo-induced dynamics of Chili–DMHBO$^+$ were studied by a combination of broadband fluorescence up-conversion and transient absorption (TA) spectroscopy with fs-time resolution. Upon excitation at 405 nm with 60 fs pulses a fluorescence band at 510 nm with $\tau < 130$ fs was found which is associated with the fluorescence of the protonated species (Fig. 4a and Supplementary Fig. 12). The peak at 560 nm in this spectrum is already due to the deprotonated chromophore in a vibronically excited state. This band is followed by an intense and broad band at 580 nm that further shifts to 600 nm with a number of time constants ranging from 0.78 to 38 ps that reflect the continuous solvent relaxation and possibly also chromophore flattening. The longest-lived band possesses a lifetime of 1.4 ns and originates from the deprotonated chromophore (Fig. 4b, c). Thus, the unexpectedly short lifetime of 130 fs is attributed to ultrafast ESPT along a pre-organized hydrogen bond. TA spectroscopy confirmed the time constants obtained by fluorescence up-conversion (Supplementary Figs. 13, 14). Fluorescence and TA measurements on Chili–DMHBI$^+$ gave a similar picture although with transient spectra shifted to shorter wavelengths (Supplementary Figs. 15–17). Again, proton transfer occurred in 120 fs. The results are consistent with proton transfer involving the strong ligand-guanine hydrogen bond observed in the crystal structure. The data also explain the fluorescence enhancement by RNA binding, as TA spectroscopy of the free DMHBO$^+$ chromophore revealed a very short lifetime of only 17 ps (Supplementary Figs. 18, 19), consistent with the very low quantum yield reported previously[8]. Moreover, the structural and mechanistic insights suggest that the fluorescent properties are dominated by the FBD, and that outside structural elements may be accessible for further engineering of the RNA fluoromodules.

**Mutagenesis and minimization of the Chili aptamer.** The exact composition and size of the stems P1 and P2 play only subordinate roles for fluorescence activation. Truncation of the basal duplex P1 (mutants P1-1, P1-2, and P1-3) retained more than 80% of the parent Chili fluorescence intensity with DMHBI$^+$ (Fig. 5a). Additional shortening of the apical hairpin resulted in the 42 nt μChili aptamer, which activated DMHBI$^+$ fluorescence to the same degree as the P1-3 mutant after overnight incubation. The junction between P1 and the FBD is formed by the Wobble

base pair U8:G45. The transition mutations U8C or G45A transformed the Wobble base pair into Watson–Crick base pairs C8:G45 or U8:A45 (mutants P1C and P1A, respectively), and resulted in slower maturation of the fluorescent complex compared to the parent Chili. Interestingly, this Wobble base pair directs the folding kinetics independent of the stem length (see G45A mutants P1A-1, P1A-2, and P1A-3). The dissociation constants for μChili or P1A-3 and DMHBI$^+$ (68 and 89 nM, respectively, Fig. 5c) were similar to the $K_D$ of the parent Chili (65 nM). Furthermore, the partially unstructured nucleotides U33-G35 are not essential for the function of the aptamer. Breaking the phosphodiester bond between U34 and G35 while connecting the original 5′ and 3′ ends of P1A-3 by a GGAA tetraloop resulted in a circularly permuted Chili aptamer (Fig. 5b) that retained 75% of the original P1A fluorescence enhancement. This finding confirms the ligand-induced folding of the quadruplex core, and suggests that the Chili aptamer may be used for fluorescent sensors[29,30] or in RNA nanotechnology[31,32].

## Discussion

The structure and fluorescence activation mechanism of the Chili aptamer is distinct from other known HBI-binding RNA aptamers like Spinach and Corn. A common feature of most structurally characterized fluorogenic aptamers is a G-quadruplex in the core of the fluorophore binding site[1], with only few exceptions[33]. The G-quartets stabilize rigid chromophore conformations by stacking interactions that largely govern the photophysical properties[34–36]. Unlike the near-planar fluorinated ligands DFHBI and DFHO in Spinach and Corn[7,20,21], the fluorophores DMHBO$^+$ and DMHBI$^+$ in Chili adopt moderately twisted conformations, resembling those found in some LSS fluorescent proteins such as LSSmKate1[19] and mCRISPred[37] (Supplementary Fig. 20 and Supplementary Table 3). In Spinach and Corn, the phenolate and imidazolone moieties of the respective ligands are embedded into a hydrogen bonding network with surrounding nucleotides and water molecules (Fig. 6). The oxime side chain of DFHO contacts the Watson–Crick face of an adenine base in the Corn RNA. In contrast, the Chili RNA stabilizes the oxime side chain of DMHBO$^+$ via H-bonding to the phosphate backbone. The G-quartet of Chili supports the benzylidene moiety of DMHBO$^+$ and the coplanar Watson–Crick base pair G15:C40, which establish the key H-bond between the phenolic hydroxy group of the ligand and the Hoogsteen edge of the Watson–Crick base pair. The central K$^+$ ion (M$_B$) also contributes to the orientation and stabilization of the ligand by including a methoxy group in the coordination sphere. Interestingly, K$^+$ can be replaced by Tl$^+$ with no change in Stokes shift, but is not easily exchanged by divalent metal ions (Mg$^{2+}$, Ba$^{2+}$, Sr$^{2+}$, and Mn$^{2+}$). For comparison, Spinach places a K$^+$ ion next to the phenolate of DFHBI, which can be replaced by Ba$^{2+}$[20].

In Chili, the key ligand-RNA hydrogen bond to the Hoogsteen edge of G15 directly controls the protonation state of the ligand in response to photoexcitation. The Chili RNA aptamer mimics the proton transfer mechanism of LSS fluorescent proteins, in which

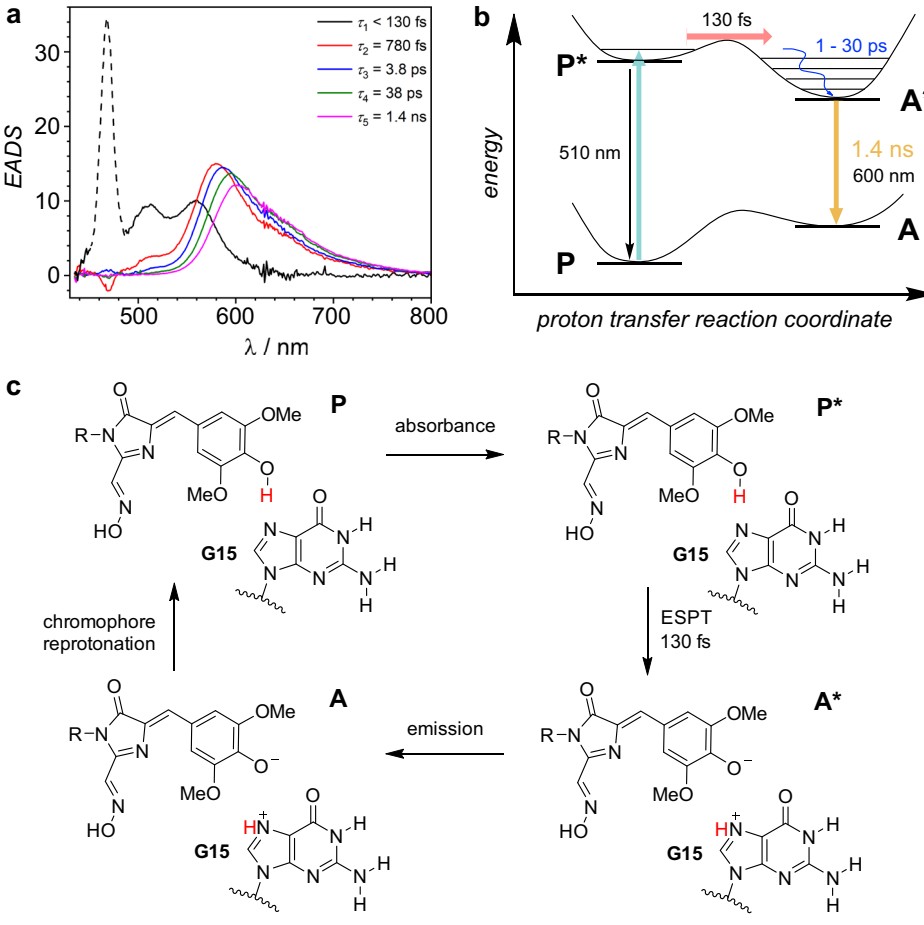

**Fig. 4 Ultrafast proton transfer in Chili-DMHBO⁺. a** Global deconvolution of time resolved fluorescence spectra from the fluorescence up-conversion measurement of Chili–DMHBO⁺. The peak at 470 nm is a Raman signal of water. The black emission spectrum (peaking at 510 nm) refers to the protonated form of DMHBO⁺, the red to magenta spectra to the deprotonated form. The time constants $\tau_2$-$\tau_4$ (0.8–38 ps) refer to a continuous dynamic Stokes shift of the emission spectrum caused by solvent and molecular relaxation. The last time constant (1.4 ns) refers to ground state recovery. **b** Schematic model of excitation, proton transfer, relaxation, and emission processes observed in (**a**). For the ESPT a barrier is assumed because of the distinct differences of the emission spectra of both states. **c** Scheme of the proposed photo cycle in Chili-DMHBO⁺. The neutral chromophore (P) is excited from the ground state. The transition from the excited phenol form (P*) to the anionic form (A*) occurs through ESPT in 130 fs. After fluorescence emission, the ground state P is regenerated.

ultrafast ESPT takes place from the chromophore to an acceptor amino acid over a short distance[19]. Our results suggest that the N7 of G15 in the Chili RNA serves as the proton acceptor. The involvement of this nitrogen atom in stabilizing the protonated ligand in the ground state was unequivocally confirmed by atomic mutagenesis. This adds an additional role to the versatile functions of guanine nucleobases found in riboswitches and ribozymes[38,39].

This work reports the study of ESPT in an RNA environment and allowed probing the RNA-ligand proton transfer dynamics in real time. The proton transfer occurred within 130 fs and is thus much faster than in wt GFP[40,41], but is comparable to what has been found for GFP mutants with short H-bonds and rewired proton transfer pathways[42–44].

Engineered fluorescent proteins harness the interplay of chromophore dynamics and protein environment to fine-tune desirable photophysical properties[45,46]. Similarly, fluctuations in RNA structures may influence fluorogenic aptamers. The folding kinetics of the Chili aptamer is strongly dependent on the U8:G45 Wobble base pair, as replacement by a Watson–Crick base pair slowed down the maturation of the fluorescent complex. An influence of quadruplex-flanking base pairs on folding and metal

ion dependence has also been reported for Spinach and Broccoli aptamers[47–49].

In summary, the structures of the Chili aptamer in complex with the green-emitting DMHBI⁺ and the orange-red-emitting DMHBO⁺ ligands unveiled the mechanism for ligand binding and fluorescence activation. The H-bond interaction between the hydroxyphenyl group and the N7 guanine nitrogen together with the coordination of a potassium ion establishes the Hoogsteen side of a G:C base pair on top of a guanine quartet as a structural motif that facilitates ESPT on the 100 fs timescale. These results further strengthen the view of G-quadruplexes flanked by canonical A-form duplexes as versatile and privileged architectures for fluorescence activation of HBI chromophores.

## Methods

**RNA synthesis and complex formation**. Unmodified RNAs were prepared by in vitro transcription with T7 RNA polymerase. The DNA templates for in vitro transcription were purchased from Sigma Aldrich or Microsynth. The reaction was performed using synthetic DNA template (1 μM), promoter strand (1 μM), and in-house T7 RNA polymerase[50]. The reaction was performed in 8 mL final volume in the presence of 40 mM Tris-HCl, pH 8.0, 30 mM MgCl₂, 10 mM DTT, 4 mM of each NTP, and 2 mM spermidine. The reaction mixture was incubated at 37 °C for 4 h, and then stopped by the addition of 100 mM EDTA pH 8.0. The RNA was

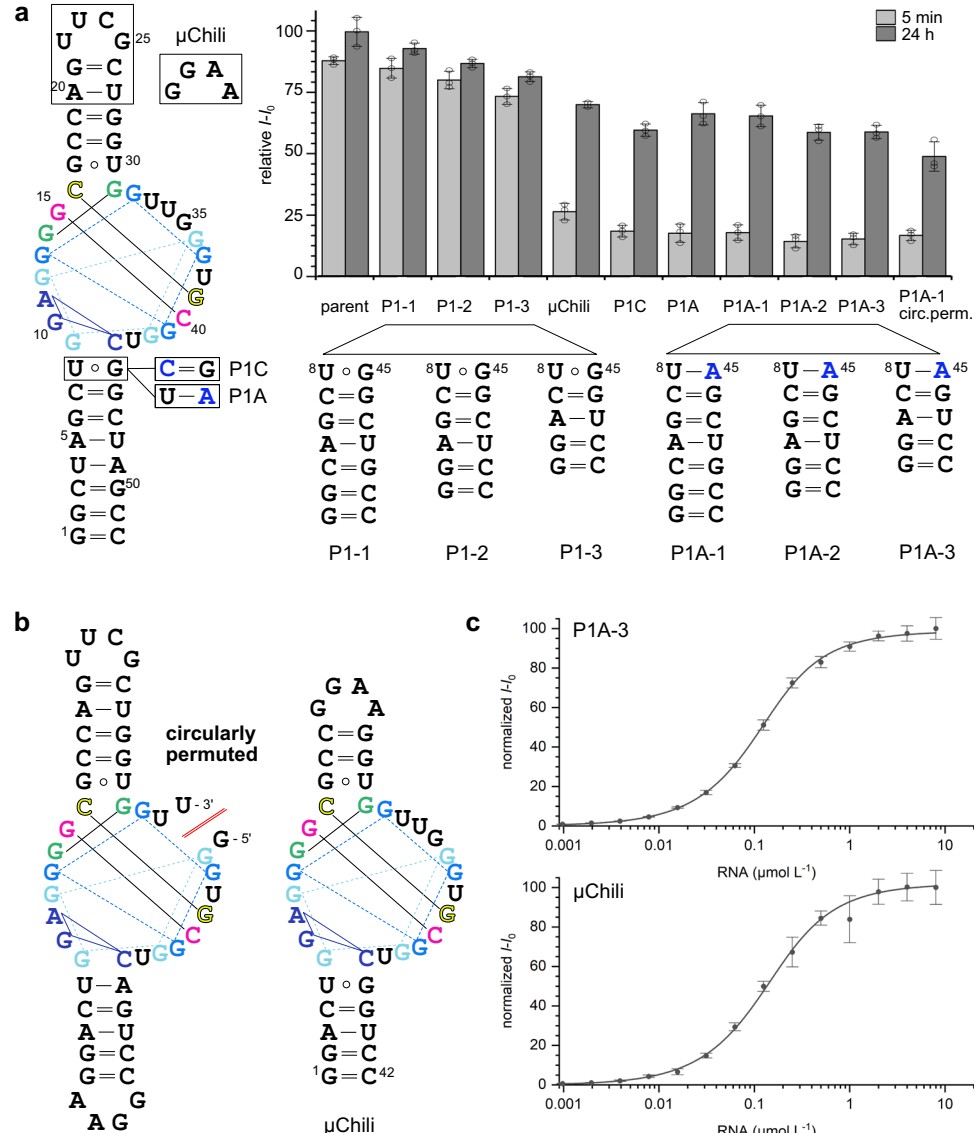

**Fig. 5 Minimization and circular permutation of the Chili aptamer. a** Mutations and truncation of the stems P1 and P2 of the parent Chili RNA are well tolerated. Fluorescence intensities of the corresponding Chili–DMHBI+ complexes, upon incubation for 5 min and 24 h, relative to the parent complex. (1 μM RNA and ligand). $n = 3$ independent experiments, mean ± SD. **b** Secondary structures of the circularly permuted Chili aptamer and μChili. **c** Titration curves of DMHBI+ (0.1 μM) with increasing RNA concentration demonstrate tight binding. $K_D$ = 68 nM for P1A-3 and 89 nM for μChili, respectively. Normalized fluorescence intensity is plotted versus RNA concentration. Data were presented as mean values ± SD ($n = 3$ independent titration experiments).

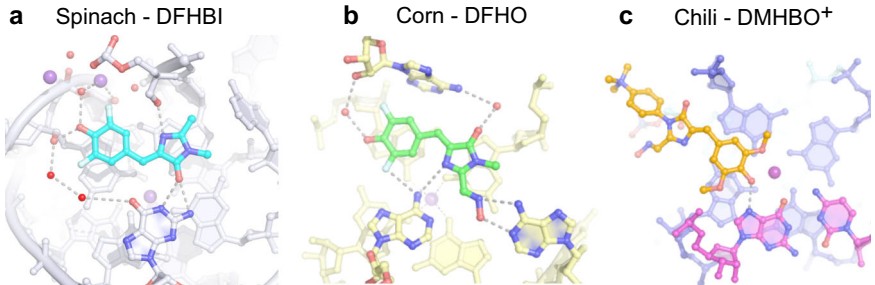

**Fig. 6 Comparison of RNA binding sites for HBI analogs in fluorogenic aptamers. a** Spinach–DFHBI (pdb 4TS0). **b** Corn–DFHO (pdb 5BJO). **c** Chili–DMHBO+ (pdb 7OAX). The ligand is shown on top of the nearest G-quartet and H-bonding interactions to the phenol/phenolate, imidazolone and oxime are indicated.

purified by 10% denaturing PAGE, extracted into TEN buffer, and recovered by precipitation with ethanol.

RNAs containing c7G and r4CI were prepared by enzymatic ligation of synthetic oligonucleotides obtained by solid phase synthesis using 2′-O-TOM-protected phosphoramidites under standard conditions[28,51]. The c7G phosphoramidite was purchased form ChemGenes, the r4CI phosphoramidite, chemical 5′ phosphorylation reagent, and 3′-phosphate CPG support were prepared as described[28]. Sequence information for RNAs are given in Supplementary Table 2.

**Crystallization of Chili DMHBI+ and DMHBO+ complexes**. Crystallization complexes were formed by mixing RNA and ligand in a 1:1.2 molar ratio in 10 mM HEPES pH 8.0, 50 mM KCl, and 1.5% DMSO. The sample was heated to 95 °C for 3 min and cooled at 23 °C for 30 min. MgCl₂ was added to a final concentration of 5 mM and the solution was stored at 4 °C for 16 h prior to setting up crystallization drops. The Chili–DMHBI+ complex was concentrated using Vivaspin 3000 MWCO ultrafiltration spin columns to ~0.5 mM and subjected to crystallization screening at 20 °C. Initial crystals appeared after 1 day. For the crystallization optimization, concentrations of the Chili–DMHBI+ (Crystal I) and Chili–DMHBO+ (Crystal II) complexes were reduced to ~0.1 mM. Larger single crystals were grown at 20 °C using the hanging and/or sitting drop vapor-diffusion method by mixing Chili–DMHBI+ and Chili–DMHBO+ complexes with solutions containing 12–17% PEG 400, 0.1 M MES pH 5.4–5.8, and 12 mM spermine tetrahydrochloride in 1:1 ratio. Drops with volumes of 0.5–2 μL produced crystals after 1–2 days and were harvested after 7 days. Prior to data collection, crystals were cryoprotected in the mother liquor containing 30% glycerol and flash frozen in liquid N₂.

Incorporation of heavy metal ions into the crystals was achieved by soaking and by co-crystallization. The co-crystallization with heavy metal was performed at 20 °C using the sitting drop vapor-diffusion method by mixing Chili–DMHBO+ complex with solutions containing 12–17% PEG 400, 0.1 M MES pH 5.4–5.8, 12 mM spermine tetrahydrochloride, and 1 mM Iridium(III) hexammine (prepared according to ref. [26]) in 1:1 ratio. Prior to data collection, Chili–DMHBO+ Iridium (III) hexammine co-crystals (Crystal III) were back-soaked in the mother liquor without Iridium(III) hexammine containing 30% glycerol and flash frozen. Soaking of the crystal was done by transferring native crystal to the mother liquor solution containing 20% glycerol and 0.1 mM Iridium(III) hexammine (Crystal IV). After 10 min, the crystal was back-soaked in mother liquor solution containing 30% glycerol and flash frozen in liquid nitrogen.

**Data collection and structure determination**. Diffraction data were collected at 100 K on EIGER16 M and PILATUS 6M detectors at the X10SA (Swiss Light Source) or P11 (DESY) beamlines. Data were indexed, integrated, and scaled with XDS[52], and reduced with POINTLESS, AIMLESS, and CTRUNCATE within the CCP4 package[53]. Initial phases were determined by single anomalous dispersion (SAD) using AutoSol Wizard in Phenix[54], with the Crystal III data collected at the Iridium L-III peak wavelength. Electron density map from AutoSol was used for automated model building with AutoBuild module in Phenix[55]. The initial model from AutoBuild was modified by iterative rebuilding in COOT[56] and refinement with Phenix.refine[57]. The partially built model from the Crystal III data were used as a MR search model in Phaser against a high resolution native dataset (Crystal II), previously corrected for anisotropy using the STARANISO server[58]. (http://staraniso.globalphasing.org/cgi-bin/staraniso.cgi). The model was completed by manual rebuilding in COOT and refined to the final $R_{free}$ value of 23.7%. The location of the DMHBO+ ligand was confirmed using both unbiased mFo-DFc and polder omit maps[59]. Ligand restraints were prepared using eLBOW in Phenix. The final structures of Crystal I and Crystal IV were solved by MR using a single 51-nt long RNA model built with Crystal II data. Water molecules were added and inspected manually. Data collection and refinement statistics are summarized in Supplementary Table 1. Structure coordinates were deposited in Protein Data Bank with the accession codes: 7OAW, 7OAX, 7OA3, and 7OAV. The structures were analyzed and figures were prepared using PyMol (Schrödinger).

**NMR spectroscopy**. RNA samples used in NMR experiments were prepared by in vitro transcription as described above. 15N labeled nucleotides were purchased from Silantes. RNA folding was achieved by thermal denaturation of the RNA at a concentration of ~0.2 mM followed by tenfold dilution with ice cold water and incubation for 1 h on ice. RNA samples were exchanged with NMR buffer (25 mM Tris-HCl, pH 7.4) using centrifuge concentrator devices (Vivaspin 3000 MWCO). RNA concentration in the final NMR samples was ~0.5 mM. The homogeneity of the folding was checked via 15% polyacrylamide native gels (acrylamide:bisacrylamide = 29:1, TBE as running buffer) at room temperature and at a power ≤1 W to avoid overheating. All the samples contained 3-(trimethylsilyl)-1-propane-sulfonic acid (DSS) as internal NMR reference and were dissolved in 10% D₂O/90% H₂O. KCl (50 mM final concentration) and 1 equivalent of DMHBI+ (ligand solution in d₆-DMSO, final concentration of d₆-DMSO ≤2% v/v) were added directly into the NMR tube to the folded RNA.

All the NMR experiments were performed on a Bruker Avance III 600 NMR spectrometer equipped with a DCH 13C/1H cryoprobe or on a Bruker Avance III 600 NMR spectrometer equipped with a BBFO room temperature probe. The

HNN-COSY experiment was performed by Dr. Helena Kovacs at Bruker Biospin (Fällanden, Switzerland) on a Bruker 700 MHz spectrometer equipped with a QCI-P cryoprobe. The spectra were acquired and processed with the software Topspin 3.2 (Bruker BioSpin, Germany). Spectra analysis was performed with the software Sparky 3.114 (Goddard, T. D.; Kneller, D. G. University of California, San Francisco). The 1H,15N-TROSY experiments were recorded using a pulse program containing the modifications proposed by Brutscher et al[60,61]. and an inter-scan delay of 0.3 s. The HNN-COSY experiment[62] was recorded using a soft WaterGATE water suppression scheme[63], an HN transfer delay of 2.5 ms, and a NN transfer delay of 15 ms.

**Steady-state UV/VIS absorption and fluorescence spectroscopy**. Steady-state fluorescence spectra were measured in Hellma ultra-micro quartz cuvettes (1.5 mm × 1.5 mm, 3 mm × 3 mm, or 10 mm × 2 mm path lengths) with a JASCO FP-8300 spectrofluorometer equipped with an FCT-817S cell changer. Steady-state UV/VIS spectra and melting curves were measured in semi-micro quartz cuvettes (10 mm path length) with a Varian Cary 100 Bio spectrophotometer equipped with a 6 × 6 Multicell Block Peltier Series II cell changer and a Varian Cary Temperature Controller.

Samples for UV/VIS and fluorescence spectroscopy were prepared with the indicated concentration of RNA, annealed by heating to 95 °C for 3 min and cooling to ambient temperature for 20 min in binding buffer (125 mM KCl, 40 mM HEPES pH 7.5)[18]. The indicated concentrations of DMHBI+ or DMHBO+ ligand and MgCl₂ (5 mM) were added. For screening of metal ion conditions, KCl was replaced by NaCl, LiCl, or TlOAc, and MgCl₂ was replaced by MnCl₂, BaCl₂, or SrCl₂. For experiments with Tl+, the RNA was precipitated from NaOAc and desalted via a centrifugal filter (Vivapsin 3000 MWCO), and MgCl₂ was replaced by Mg(OAc)₂.

Fluorescence spectra of the samples were measured 5 min after addition of the ligand and again after an additional incubation period of 24 h at 4 °C. Samples were excited at 413 nm (when using DMHBI+) or 456 nm (when using DMHBO+) and fluorescence emission spectra recorded up to 750 nm. Other instrument parameters (Ex/Em slit widths, scan speed, integration time, and detector sensitivity) were kept constant.

For binding affinity titrations, fluorescence spectra were measured 24 h after preparation of the individual samples and the emission bands were integrated. The resulting values (mean ± SD for n = 3) were fitted to a one-site binding model in OriginPro 2019[8].

For melting curve measurements, 5 ramps between 10 and 95 °C (UV: 0.5 °C/min, fluorescence: 1 °C/min or 5 °C/min) were recorded, the first ramp served for annealing and was not included in the data analysis. The samples were layered with silicon oil to minimize evaporation during the experiment.

**Time-resolved spectroscopy**. TCSPC measurements were performed on a Horiba DeltaFlex system with a DeltaDiode DD-320 excitation source (time calibration: 0.026 ns/channel, 4096 channels). The samples were prepared as described above at an RNA concentration of 0.5 and 1 μM ligand (in 125 mM KCl, 40 mM HEPES pH 7.5, and 5 mM MgCl₂). A sample without RNA and ligand was used to measure the instrument response function (Em wavelength = Ex wavelength). Data analysis was performed with Data Analysis Station 6 (Horiba) by fitting to an appropriate sum of exponentials. Fit quality was judged by the reduced $\chi^2$ value (<1.2 without improvement due to additional exponential terms) and a random distribution of the residuals.

**Broadband fluorescence upconversion (FLUPS) and transient absorption spectroscopy (TAS)**. All measurements were performed in 1 or 0.2 mm quartz cuvettes. The samples were prepared as described above at a complex concentration of 250 or 625 μM in 125 mM KCl, 40 mM HEPES pH 7.5, and 5 mM MgCl₂. Both spectrometers, the broadband fluorescence upconversion, and the transient absorption spectrometer, were driven by a chirped pulse amplified femtosecond laser "Solstice" from Newport-Spectra with a fundamental wavelength of 800 nm, a pulse width of 100 fs, and a repetition rate of 1 kHz.

The FLUPS is a commercial available spectrometer from LIOPTEC which can simultaneously measure 395–850 nm fluorescence with an intrinsic resolution of 0.9 nm (303–516 nm upconverted, intrinsic resolution 0.42 nm). The maximum delay between pump and gate pulse can be up to 1.500 ps. To generate the pump wavelength 405 nm for the FLUPS we used the output from a non collinear parametric amplifier which was set to 810 nm. The 810 nm were guided through a BBO crystal to generate the second harmonic at 405 nm. We used an 800 nm broadband bandpass filter to eliminate the residual 810 nm. The pump pulse energy was 50 nJ and the gate pulse energy was 30 uJ at 1280 nm. For more details on the setup see ref. [64].

The pump pulse from the TAS was generated with a traveling wave oscillating parametric amplifier "TOPAS-C" from Light Conversion. The pump pulse energy was set to 40 nJ. The white light was generated with a moving CaF₂ crystal with some filters to achieve a probe light from 400 to 800 nm. The maximum delay between pump and probe was 8 ns. For a more detailed description of the setup see ref. [65].

For both types of measurement we performed a global analysis of the data using GLOTARAN software[66]. This analysis takes the white light dispersion and, in case of the TA measurements, the coherent artifact into account and yields evolution associated difference spectra (EADS) for a consecutive kinetic model and decay associated difference spectra (DADS) for a parallel kinetic model[67].

**Reporting summary**. Further information on experimental design is available in the Nature Research Reporting Summary linked to this paper.

## Data availability

Structural data obtained by X-ray crystallography were deposited in the Protein Data Bank (PDB) and are available with the following accession codes: 7OAW, 7OAX, 7OA3, 7OAV. All relevant data are provided in the Figures and Supplementary Information (Supplementary Figs. 1–20). Publicly available datasets used in this study: PDB 3NT9, 4TS0, 5BJO, 6XWY. Source data are provided with this paper.

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

## Acknowledgements

This work was supported by the European Research Council (ERC/No 682586), the Deutsche Forschungsgemeinschaft (DFG), the Max Planck Society, the University of Würzburg, and the SolTech Initiative of the Bavarian State Ministry of Education, Culture, Science, and the Arts. We thank Ulrich Steuerwald and Jürgen Wawrzinek for technical assistance in the crystallization facility (MPI for biophysical chemistry), Matthias Grüne (University Würzburg) and Helena Kovacs (Bruker BioSpin AG) for NMR support, Matthias Stolte (University Würzburg) for the fluorescent images of crystals, and the beamline staff at DESY (PETRA III, P11) and at the Swiss Light Source PSI (PXII) for assistance with data collection.

## Author contributions

M.M., C.S., I.B., and C.H. designed the study, M.M. crystallized the complexes and solved the structure under the supervision of V.P.; TA and FLUPS experiments were performed by A.S. and analysed by M.H. and C.L.; all other experiment were performed by C.S., I.B., A.-K.L. and C.H. All authors contributed to writing the manuscript.

## Funding

## Competing interests

The authors declare no competing interests.
