## [Peer Review File · Nature Communications]

REVIEWER COMMENTS

Reviewer #1 (Remarks to the Author):

In this manuscript, the Lambert, Pena, and Höbartner groups formed a collaboration to understand the structural mechanism by which the Chili aptamer attains a large Stokes shift. In this study the groups determined two crystal structures of Chili bound to the HBI derivatives DMHBI⁺ and DMHBO⁺, used NMR spectroscopy to determine the proton acceptor, and fluorescence up-conversion to determine the rate of proton transfer to the acceptor N7 guanine. Generally speaking, this work is very interesting to both a broad and specialist community, is well thought out, and well executed. Before I continue with my critique of this work I would like to unambiguously state my support publication for this manuscript in Nature Communications. However, there are a number of major and minor concerns I would like to see addressed prior to full acceptance.

Major:

1) The authors use the abbreviation ca. in a number of instances. This is rather unconventional for scientific writing and the exact values should be inserted (in most cases) with a calculation of error proceeding the value.

2) While there are a number of times in the manuscript that metal ion M(B) is mentioned, there is rather little experimentation or discussion as to the functional significance of this group. In my opinion, this is the most interesting aspect of the Chili structures. What role does this metal ion play in inducing a large Stokes shift? Does this metal alter the rate of proton transfer? There are a number of experiments that I can imagine addressing these concerns. At a minimum, I would like to see the authors measure the Stokes shift of Chili-DFHBO⁺ and Chili-DMHBI⁺ in the presence of cations other than K⁺ that stabilize G-quadruplex formation. It should be possible to prefold Chili in K⁺ and exchange out K⁺ for the ions Sr²⁺, Ba²⁺, and Tl⁺ (note Tl⁺ is highly toxic and forms insoluble aggregates with Cl⁻. Remove all chloride from solution prior to adding Tl⁺). Coordination of these metal ions via the G-quadruplex-ligand interface should alter the H-bond length to G15 and possibly alter electronic coupling of the ligand-RNA complex.

The crystal structure of Spinach-DFHBI (PDB ID: 4TS0) places a similar fluorophore phenolate adjacent to a K⁺. Please compare your results with this structure in Figure 6 as it is more relevant to your discussion.

3) Lines 236-239. The twisted fluorophore conformations observed in Chili is compared to LSSmKate and mCRISPred. Are you suggesting that imposing a twisted fluorophore induces LSS? If so, please be more direct with this statement and try to provide further support. The structure of the Mango-I aptamer demonstrated that a twisted fluorophore reduced quantum yield without change in Stokes shift.

4) In Figure 1b bases G9-C44 along with C16-G31 should be shown as base-paired. Figure 1b and d do not show the appropriate "Leontis-Westhof" symbols for the GU wobble. These are cis not trans pairs. There should be a filled circle.

5) Given there are a number of times bond distances were discussed throughout the manuscript, the Maximum-likelihood coordinate precision of the structures should be reported in the crystallographic table. This will allow the reader to establish significance of the statements regarding bond lengths. Also, Supplementary Table 1 should be rechecked for significant digits reported. I find it highly unlikely that the unit cell dimensions are significant out to the second decimal place at this resolution.

Minor:

There are a number of typographical errors and issues with writing composition throughout.

- 1) line 49: "side chain" is not the correct term in this instance.
- 2) The section "Overall structure of the Chili RNA aptamer bound to DMHBO⁺ and DMHBI⁺" contains a number of statements that I would expect in a Methods section, not results.
- 3) Line 80: "High-resolution" should be removed. Traditionally in the crystallography community high resolution would start around 1.5 Å at best. I would call this moderate resolution.
- 4) Line 83: "the native dataset contained four copies of RNA-ligand complexes". Which native data? There are two reported here. Please be specific throughout and report appropriately.
- 5) Line 86: "comparable" should be replaced by "similar".
- 6) Lines 130-132 are redundant with previous statement.
- 7) Lines 139-140: A difference in KD within an order of magnitude is not "striking".
- 8) Figure 2: Please be consistent with how structures are rendered. Figure 2f is not ball-stick/cartoon like rest of panels.
- 9) Figure 4: Please remove the "7" by the G15 N7s. In some cases it appears like a net charge of +7.
- 10) As mentioned previously, I think Figure 6a should be replaced with the Spinach-DFHBI structure 4TS0 with K⁺ shown. The current panel has PDB ID 5OB3 which is iSpinach-DFHBI, not Spinach-DFHBI.

Reviewer #2 (Remarks to the Author):

Overall, this is an interesting manuscript which primarily focuses on the structure of an RNA aptamer called Chili. This aptamer was developed by the Hobartner laboratory, along with a specific fluorophore, which when they interact to form a very far-red shifted fluorogenic aptamer complex. This manuscript adds to the recent set of structures fluorogenic aptamers and their bound ligands. Notably, there are lots of interesting surprises and insights that are derived from this fluorogenic aptamer and its bound fluorophore. The authors provide new mechanistic insight into how this aptamer achieves its large Stokes shift, showing a type of excited state proton transfer mechanism. The authors use very thoughtful approaches to prove specific binding and fluorescence mechanisms involving modified nucleotides at specific locations informed by the structure. Overall, I think the manuscript is well done, very interesting, well written, and properly supported by all the presented data.

I have some comments, all of which are fairly minor:

1. I thought the discussion of the heating and cooling was unclear in the text. Only after thinking about this and looking at the figure, did it become clear to me that I think the authors are using the heating to melt the RNA and thus liberate the fluorophore, allowing it to switch to the standard solution-phase phenolate form. I would suggest that they look at how they wrote the description between lines 155 and 159. RNA melting is not even mentioned in this section, but I think this is the underlying rationale for the different temperatures that are being used here.
2. I was also confused about the discussion between lines 164 and 168. They bring up a supposedly interesting proton chemical shift, but I'm not sure how this relates to the previous or subsequent paragraph. Does the shift have anything to do with G15? I think this little paragraph needs to be rewritten so it is properly connected with the subsequent paragraph, to which I think it is related.
3. It would be interesting if the authors can mention a little bit about the extremely fast proton transfer of 130 fs might have significance or utility especially compared to fluorescent proteins which are much slower. How could this be beneficial? What specific applications would the specific excited state proton transfer rates described here enable? I think this manuscript could benefit a little bit more about explaining in a more concrete way how the specific findings described here enable new types of imaging or new types of sensor technologies.
4. One of the things that I found most interesting was figure 6, which does a side-by-side comparison of the crystal structures of Spinach, Corn and Chili. I wish the authors had been comparing and showing these comparisons throughout the manuscript rather than putting it at the very end, and

including only discussion in the Discussion section. To me, a deeper comparison of how the different structures result in different fluorescence properties would have been very beneficial.

Nevertheless, this is a very solid manuscript. The authors have done a good job bringing together structure and ultrafast proton transfer measurements (which to the best my knowledge have not been performed in the fluorogenic aptamer field) to provide insights into this interesting fluorogenic aptamer.

Reviewer #3 (Remarks to the Author):

This manuscript reported a comprehensive study of the crystal structures, site-directed mutagenesis, and various spectroscopies of Chili RNA with the positively charged DMHBO⁺ and DMHBI⁺, which display fluorogenic properties. The main text was clearly and concisely written with much information well organized in the SI. This reviewer would suggest the authors to consider the following points for minor revision.

1. What occurs to the -C=N-OH group (other than the phenolic hydroxyl group) after photoexcitation of the chromophore? If the authors want to exclude its potential contribution inside the FBD after photoexcitation, pKa determination and other characterizations (structural constraints) should be clearly given.
2. Since ultrafast electronic spectroscopy could help uncover the ~130 fs ESPT process inside FBD, it is useful to perform similar experiments on the chromophore in solution (neutral and basic pH) to confirm the spectral signatures of the protonated and deprotonated forms.
3. Different colors for base pairs should be defined in Fig. 1 caption.
4. Ultrafast spectroscopy typically uses nanometer unit for the excitation and emission, as well as pump and probe. Figs. S9-S16 captions and units should be revised accordingly. The units for "EADS" and "DADS" should also be given for the vertical axes in Figs. S9, S10, S12, S13, S15. The color-coded time-resolved spectra are unlabeled versus time delay in Figs. S11, S14, S16, it is difficult to tell the spectral change as time progresses.
5. What is the fluorescence quantum yield of the Chili-DMHBO⁺ and Chili-DMHBI⁺ in free form versus bound with RNA aptamer (complex form)? What about other nonradiative energy relaxation pathways in Fig. 4b (especially since the chromophore flattening in the excited state was mentioned on p. 9)?
6. Wording issues, for example, "to elicits" in Abstract, "..." is unusual for authors in references (e.g., ref. 19-20 in SI), "pdb xxxx" in Fig. 6 caption and Table S3.

Point-by-point reply to the referees' comments on manuscript NCOMMS-21-07329-T

We thank the reviewers for their enthusiastic and constructive comments on our manuscript. We address all the queries of the three reviewers as outlined below. The corresponding changes in the manuscript are marked in **red**. Our response to the reviewer's queries is marked in **blue**.

Reviewer #1 (Remarks to the Author):

In this manuscript, the Lambert, Pena, and Höbartner groups formed a collaboration to understand the structural mechanism by which the Chili aptamer attains a large Stokes shift. In this study the groups determined two crystal structures of Chili bound to the HBI derivatives DMHBI⁺ and DMHBO⁺, used NMR spectroscopy to determine the proton acceptor, and fluorescence up-conversion to determine the rate of proton transfer to the acceptor N7 guanine. Generally speaking, this work is very interesting to both a broad and specialist community, is well thought out, and well executed. Before I continue with my critique of this work I would like to unambiguously state my support publication for this manuscript in Nature Communications.

However, there are a number of major and minor concerns I would like to see addressed prior to full acceptance.

Major: 1) The authors use the abbreviation *ca.* in a number of instances. This is rather unconventional for scientific writing and the exact values should be inserted (in most cases) with a calculation of error proceeding the value.

Reply: The abbreviation *ca.* has been replaced by appropriate values.

2) While there are a number of times in the manuscript that metal ion M(B) is mentioned, there is rather little experimentation or discussion as to the functional significance of this group. In my opinion, this is the most interesting aspect of the Chili structures. What role does this metal ion play in inducing a large Stokes shift? Does this metal alter the rate of proton transfer? There are a number of experiments that I can imagine addressing these concerns. At a minimum, I would like to see the authors measure the Stokes shift of Chili-DFHBO⁺ and Chili-DMHBI⁺ in the presence of cations other than K⁺ that stabilize G-quadruplex formation. It should be possible to prefold Chili in K⁺ and exchange out K⁺ for the ions Sr²⁺, Ba²⁺, and Tl⁺ (note Tl⁺ is highly toxic and forms insoluble aggregates with Cl⁻. Remove all chloride from solution prior to adding Tl⁺). Coordination of these metal ions via the G-quadruplex-ligand interface should alter the H-bond length to G15 and possibly alter electronic coupling of the ligand-RNA complex.

Reply: We agree that metal ion M(B) is of special interest. As we reported earlier, Mg²⁺ is not essential for Chili folding and fluorescence activation, since 125 mM K⁺ alone gave 80% of the maximum fluorescence intensity. Moreover, the addition of Ba²⁺ did not induce a shift in the emission maximum of Chili-DMHBI⁺ when the RNA was folded in K⁺-buffer, but reduced the intensity (Steinmetzger et al, Nucleic Acids Res 2019). We also co-crystallized the Chili-DMHBO⁺ complex with Mn²⁺ ions, but the anomalous signal was weak, supporting the assignment of M(B) as K⁺, which was not easily replaced by divalent metal ions.

We thank the reviewer for the suggestion to study the functional significance of M(B) in more detail using potassium surrogates. Interesting new results are included and the data are shown in the new Supplementary Figures 7 and 8.

We found that K⁺ can be fully replaced by Tl⁺ in Chili-DMHBO⁺, with no change in Stokes shift and emission intensity. We are not aware of other studies of fluorogenic RNA aptamers using Tl⁺ to draw a comparison. However, earlier studies on DNA quadruplexes by NMR and crystallography had demonstrated the ability of Tl⁺ to closely mimic K⁺ in nucleic acids.

On the other hand, replacing K^+ by Na^+ was not necessarily expected to maintain the same G-quadruplex structure. Na^+ has a smaller ionic radius and has been reported to prefer binding to a single G-quartet plane, rather than between two adjacent G-quartet planes as usually seen for K^+ .

Nevertheless, we also observed fluorescence activation in Chili-DMHBO⁺ when K^+ was replaced by Na^+ , although with only 25% residual intensity, while the Stokes shift remained unaffected. These data suggest that Na^+ may be more easily exchangeable than K^+ , which could allow binding of a divalent metal ion in place of M(B) and change the electronic coupling of the ligand-RNA complexes. Indeed, for both Chili-DMHBO⁺ and Chili-DMHBI⁺, the addition of Sr^{2+} or Ba^{2+} resulted in a bathochromic shift of the excitation and emission maxima only in the presence of Na^+ , but not in the presence of K^+ . Full replacement of K^+ by Sr^{2+} or Ba^{2+} did not result in an active aptamer fold.

The crystal structure of Spinach-DFHBI (PDB ID: 4TS0) places a similar fluorophore phenolate adjacent to a K^+ . Please compare your results with this structure in Figure 6 as it is more relevant to your discussion.

Reply: We have replaced Figure 6a with the Spinach-DFHBI structure of pdb 4TS0 and discussed the reported structure with respect to the K^+ ion.

3) Lines 236-239. The twisted fluorophore conformations observed in Chili is compared to LSSmKate and mCRISPred. Are you suggesting that imposing a twisted fluorophore induces LSS? If so, please be more direct with this statement and try to provide further support. The structure of the Mango-I aptamer demonstrated that a twisted fluorophore reduced quantum yield without change in Stokes shift.

Reply: No, our statement was not meant to imply that the twisted chromophore conformation is responsible for the large Stokes shift. Also in LSS FPs, the large Stokes shift is associated with ESPT, and not with the chromophore twist. Our data and statement are in line with the reviewer's comment and with the findings on the Mango-I aptamer (which does not involve proton transfer pathways).

4) In Figure 1b bases G9-C44 along with C16-G31 should be shown as base-paired. Figure 1b and d do not show the appropriate "Leontis-Westhof" symbols for the GU wobble. These are cis not trans pairs. There should be a filled circle.

Reply: The two base pairs mentioned by the reviewer do not exist, therefore cannot be shown as base-paired in Figure 1b. In short, G9 is part of the G-tetrad T1 (cyan), C44 belongs to the GAC base triple (dark blue). C16 forms a Watson-Crick base pair with G39 (yellow), and G31 is the H-bonding partner of G14 (green). We would also like to refer to the chosen color code and the diagram in Fig 1d, shown side-by-side to appreciate the fascinating architecture of the Chili core.

Regarding Leontis-Westhof symbols, we thank the reviewer for catching the inconsistency. The correct filled symbol has been inserted.

5) Given there are a number of times bond distances were discussed throughout the manuscript, the Maximum-likelihood coordinate precision of the structures should be reported in the crystallographic table. This will allow the reader to establish significance of the statements regarding bond lengths. Also, Supplementary Table 1 should be rechecked for significant digits reported. I find it highly unlikely that the unit cell dimensions are significant out to the second decimal place at this resolution.

Reply: The Maximum-likelihood coordinate precision has been added to the crystallographic table and the significant digits corrected.

Minor:

There are a number of typographical errors and issues with writing composition throughout.
1) line 49: "side chain" is not the correct term in this instance.

Reply: The term "side chain" has been replaced by "group".

2) The section "Overall structure of the Chili RNA aptamer bound to DMHBO+ and DMHBI+" contains a number of statements that I would expect in a Methods section, not results.

Reply: We have shortened this paragraph accordingly.

3) Line 80: "High-resolution" should be removed. Traditionally in the crystallography community high resolution would start around 1.5 Å at best. I would call this moderate resolution.

Reply: Agreed. The term has been removed.

4) Line 83: "the native dataset contained four copies of RNA-ligand complexes". Which native data? There are two reported here. Please be specific throughout and report appropriately.

Reply: We have checked for consistency and updated where necessary. The native datasets of both Chili-DMHBI⁺ (crystal I) and Chili-DMHBO⁺ (crystal II) each contained four copies of the RNA-ligand complexes.

5) Line 86: "comparable" should be replaced by "similar".

Reply: agreed and replaced.

6) Lines 130-132 are redundant with previous statement.

Reply: We agree that G:G trans sugar-sugar edge (tSS) should be sufficient, but for readers not so fluent on RNA structure nomenclature, we prefer to spell out the interactions that form this non-canonical base pair.

7) Lines 139-140: A difference in KD within an order of magnitude is not "striking".

Reply: agreed and deleted.

8) Figure 2: Please be consistent with how structures are rendered. Figure 2f is not ball-stick/cartoon like rest of panels.

Reply: the rendering of Figure 2f has been adjusted to fit the style of the other figures.

9) Figure 4: Please remove the "7" by the G15 N7s. In some cases it appears like a net charge of +7.

Reply: The figure has been updated to avoid any confusion.

10) As mentioned previously, I think Figure 6a should be replaced with the Spinach-DFHBI structure 4TS0 with K⁺ shown. The current panel has PDB ID 5OB3 which is iSpinach-DFHBI, not Spinach-DFHBI.

Reply: agreed and replaced.

Reviewer #2 (Remarks to the Author):

Overall, this is an interesting manuscript which primarily focuses on the structure of an RNA aptamer called Chili. This aptamer was developed by the Hobartner laboratory, along with a specific fluorophore, which when they interact to form a very far-red shifted fluorogenic aptamer complex. This manuscript adds to the recent set of structures fluorogenic aptamers and their bound ligands. Notably, there are lots of interesting surprises and insights that are derived from this fluorogenic aptamer and its bound fluorophore. The authors provide new mechanistic insight into how this aptamer achieves its large Stokes shift, showing a type of excited state proton transfer mechanism. The authors use very thoughtful approaches to prove specific binding and fluorescence mechanisms involving modified nucleotides at specific locations informed by the structure. Overall, I think the manuscript is well done, very interesting, well written, and properly supported by all the presented data.

I have some comments, all of which are fairly minor:

1. I thought the discussion of the heating and cooling was unclear in the text. Only after thinking about this and looking at the figure, did it become clear to me that I think the authors are using the heating to melt the RNA and thus liberate the fluorophore, allowing it to switch to the standard solution-phase phenolate form. I would suggest that they look at how they wrote the description between lines 155 and 159. RNA melting is not even mentioned in this section, but I think this is the underlying rationale for the different temperatures that are being used here.

Reply: Thanks for pointing this out. We have changed the wording and added the statement on RNA melting.

2. I was also confused about the discussion between lines 164 and 168. They bring up a supposedly interesting proton chemical shift, but I'm not sure how this relates to the previous or subsequent paragraph. Does the shift have anything to do with G15? I think this little paragraph needs to be rewritten so it is properly connected with the subsequent paragraph, to which I think it is related.

Reply: Correct, the text relates to the G15-C40 base pair. By including the new data on various metal ions (see above response to reviewer 1), we have now better connected this paragraph in the context of the functional importance of G15.

3. It would be interesting if the authors can mention a little bit about the extremely fast proton transfer of 130 fs might have significance or utility especially compared to fluorescent proteins which are much slower. How could this be beneficial? What specific applications would the specific excited state proton transfer rates described here enable? I think this manuscript could benefit a little bit more about explaining in a more concrete way how the specific findings described here enable new types of imaging or new types of sensor technologies.

Reply: The major benefit of the Chili fluoromodule for imaging and sensor applications lies in the large Stokes shift, rather than the direct readout of the fast proton transfer rate. Several fluorescent proteins with similarly fast proton transfer rates are known. Large Stokes shifts enable fluorophore multiplexing, such as dual color monitoring by excitation of two different FPs at the same wavelength, and the development of FRET pairs with minimal spectral bleed-through. For fluorescent proteins, multiplexing of regular FPs and LSS FPs is well established, and can for example be used for FRET-FLIM imaging. We propose that Chili fills this gap in the toolbox of fluorogenic aptamers, and may also serve as partner for multiplexing with regular aptamer fluoromodules.

4. One of the things that I found most interesting was figure 6, which does a side-by-side comparison of the crystal structures of Spinach, Corn and Chili. I wish the authors had been comparing and showing these comparisons throughout the manuscript rather than putting it at the very end, and including

only discussion in the Discussion section. To me, a deeper comparison of how the different structures result in different fluorescence properties would have been very beneficial.

Reply: We agree that the comparison in Fig 6 is important, and have update it according to the suggestion by reviewer 1. The comparison to previously reported aptamer structures throughout the text is not easily feasible due to length restrictions, and it would distract from the flow of information on the description of new results. We believe the side-by-side comparison is best placed in the discussion section and we have expanded it to also reflect the new metal ion data in comparison to Spinach.

Nevertheless, this is a very solid manuscript. The authors have done a good job bringing together structure and ultrafast proton transfer measurements (which to the best my knowledge have not been performed in the fluorogenic aptamer field) to provide insights into this interesting fluorogenic aptamer.

Reviewer #3 (Remarks to the Author):

This manuscript reported a comprehensive study of the crystal structures, site-directed mutagenesis, and various spectroscopies of Chili RNA with the positively charged DMHBO⁺ and DMHBI⁺, which display fluorogenic properties. The main text was clearly and concisely written with much information well organized in the SI. This reviewer would suggest the authors to consider the following points for minor revision.

1. What occurs to the -C=N-OH group (other than the phenolic hydroxyl group) after photoexcitation of the chromophore? If the authors want to exclude its potential contribution inside the FBD after photoexcitation, pKa determination and other characterizations (structural constraints) should be clearly given.

Reply: The -C=N-OH group is anchored by two hydrogen bonds to the phosphate backbone of the RNA at G10 and to the 2'-OH group of G9. These structural constraints are depicted in Figure 2f in the manuscript. The UV-Vis pKa titration of DMHBO⁺ between pH 1 and pH 12 revealed pKa 6.9 for the phenol and pKa 9.2 for the oxime. The spectra and titration curves are shown in the new Supplementary Figure 6.

2. Since ultrafast electronic spectroscopy could help uncover the ~130 fs ESPT process inside FBD, it is useful to perform similar experiments on the chromophore in solution (neutral and basic pH) to confirm the spectral signatures of the protonated and deprotonated forms.

Reply: Transient absorption spectroscopy has been performed on the free ligand DMHBO⁺ at pH 7.5. At this pH both protonated and deprotonated forms are populated, and the signatures can be seen in Supplementary Figures 18 and 19.

3. Different colors for base pairs should be defined in Fig. 1 caption.

Reply: The color code has been added.

4. Ultrafast spectroscopy typically uses nanometer unit for the excitation and emission, as well as pump and probe. Figs. S9-S16 captions and units should be revised accordingly. The units for "EADS" and "DADS" should also be given for the vertical axes in Figs. S9, S10, S12, S13, S15. The color-coded time-resolved spectra are unlabeled versus time delay in Figs. S11, S14, S16, it is difficult to tell the spectral change as time progresses.

Reply: Different disciplines and laboratories have different preferences in the use of wavelength in nm or wavenumbers in cm^{-1} . For this reason, the submitted Supplementary Figures S9-16 showed both units, wavenumbers on the bottom and wavelength on the top x-axis. For consistency with the other Figures in the SI, wavelengths are now shown on the bottom axis and added to the Figure captions (Supplementary Figures S13-S19 (new numbers)). EADS and DADS are obtained from dimensionless optical density, and therefore only have arbitrary units. The color code in each panel of figures S11,14,16 (new numbers S14,17,19) is defined from blue to red with increasing time. This information has been added to the figure captions.

5. What is the fluorescence quantum yield of the Chili-DMHBO⁺ and Chili-DMHBI⁺ in free form versus bound with RNA aptamer (complex form)? What about other nonradiative energy relaxation pathways in Fig. 4b (especially since the chromophore flattening in the excited state was mentioned on p. 9)?

Reply: The quantum yields for Chili-DMHBO⁺ and Chili-DMHBI⁺ were reported in Steinmetzger et al. *Chem. Eur. J.* 2019., and are given here for reference. Quantum yields of DMHBI⁺: free 0.001, bound: 0.4; DMHBO⁺: free 0.001, bound: 0.094. Additional non-radiative energy relaxation pathways are likely involved, but are not depicted in the simplified scheme of Figure 4b.

6. Wording issues, for example, “to elicits” in Abstract, “...” is unusual for authors in references (e.g., ref. 19-20 in SI), “pdb xxxx” in Fig. 6 caption and Table S3.

Reply: Thanks for pointing these out. The issues have been corrected.

REVIEWER COMMENTS

Reviewer #1 (Remarks to the Author):

I like the revisions that the authors have made. I'm very pleased to see that the fluorescence measurements in different cations yielded interesting results. I suggest to publish as is.

Reviewer #3 (Remarks to the Author):

The authors have made the proper revisions following the reviewer comments. I have no further concerns for the current presentation of major findings. Minor typos such as "these results suggests", "proton transfer occurred" will be corrected later.

Reply to the referees' comments on manuscript NCOMMS-21-07329A

We thank the reviewers for carefully reading our revised manuscript.

Reviewer #1 (Remarks to the Author):

I like the revisions that the authors have made. I'm very pleased to see that the fluorescence measurements in different cations yielded interesting results. I suggest to publish as is.

We thank the reviewer for the kind comments. No further changes were suggested.

Reviewer #3 (Remarks to the Author):

The authors have made the proper revisions following the reviewer comments. I have no further concerns for the current presentation of major findings. Minor typos such as “these results suggests”, “proton transfer occurred” will be corrected later.

We thank the reviewer for the support and for pointing out the minor typos, which have now been corrected.